# DAA Treatment Failure in a HIV/HBV/HCV Co-Infected Patient Carrying a Chimeric HCV Genotype 4/1b

**DOI:** 10.3390/ijerph191811655

**Published:** 2022-09-15

**Authors:** Maria Antonia De Francesco, Franco Gargiulo, Serena Zaltron, Angiola Spinetti, Francesco Castelli, Arnaldo Caruso

**Affiliations:** 1Institute of Microbiology, Department of Molecular and Translational Medicine, University of Brescia ASST Spedali Civili, 25123 Brescia, Italy; 2Division of Infectious and Tropical Diseases, Department of Clinical and Experimental Sciences, University of Brescia and ASST Spedali Civili, 25123 Brescia, Italy

**Keywords:** SVR, DAA, recombination, genotype

## Abstract

Approved direct antiviral agent (DAA) combinations are associated with high rates of sustained virological response (SVR) and the absence of a detectable hepatitis C viral load 12–24 weeks after treatment discontinuation. However, a low percentage of individuals fail DAA therapy. Here, we report the case of a HIV/HBV/HCV co-infected patient who failed to respond to DAA pangenotypic combination therapy. The sequencing of NS5a, NS5b, NS3 and core regions evidenced a recombinant intergenotypic strain 4/1b with a recombination crossover point located inside the NS3 region. The identification of this natural recombinant virus underlines the concept that HCV recombination, even if it occurs rarely, may play a key role in the virus fitness and evolution.

## 1. Introduction

Hepatitis C virus (HCV) is a blood-borne virus that often leads to the development of liver cirrhosis and hepatocellular carcinoma in chronically infected patients. The World Health Organization (WHO) estimated that about 58 million people are affected by chronic HCV infection, with 1.5 million new infections occurring per year [1]. To date, eight genotypes and 86 subtypes with a sequence divergence of >15% have been identified [2,3,4].

Although HCV is characterized by a high mutation rate, recombination is a rare event. However, different intergenotypic, called chimeras, and intragenotypic recombinant HCV strains have been identified around the world [5,6,7,8,9,10,11].

HCV genotyping is generally performed through assays targeting only the 5′ UTR (untranslated region), or core regions, which fail to identify chimeric HCV strains.

In recent years, the treatment of chronic HCV infection improved with the introduction of direct antiviral agents (DAAs), which were able to achieve high rates of sustained virological response (SVR). The increasing availability of pangenotypic DAA combinations such as sofosbuvir/velpatasvir and glecaprevir/pibrentasvir has simplified HCV therapy without requiring the knowledge of the HCV genotype and subtype.

However, 2–3% of patients treated with these combinations do not achieve SVR. The reasons for this failure, include the emergence of resistance associated variants (RAVs), potential HIV coinfection [12] and difficult to treat some HCV genotypes or possibly chimeric HCV strains.

Here we report the case of pangenotypic DAA treatment failure in a HIV/HBV/HCV co-infected patient carrying a recombinant 4/1b genotype.

## 2. Case

In 2006, a 55-year-old man attended the Division of Infectious and Tropical Diseases of Spedali Civili’s Hospital, Brescia, Italy for a chronic HCV infection diagnosed in 1992. He was a drug injection user and acquired HCV by parenteral route. There was no history of alcohol abuse. However, his medical history reported an occult HBV infection (OBI) and laboratory results showed that the patient was negative for hepatitis B surface (HBs) antigen and HBV DNA but was positive for both anti-hepatitis B core antibody and anti-HBs antibody. 

At that time, the patient was not treated against HCV due to an interferon-alpha contraindication because he suffered from psychosis and a mood disorder.

The HCV genotype was determined as 4a/4c/4d by using LiPA test targeting 5′ UTR and core regions (Versant HCV genotype 2.0 Assay LiPA; Siemens Healthcare Diagnostics). A mild fibrosis with a F1/F2 score was determined by transient elastography (FibroScan). Ultrasound scans did not show hepatic foci, portal hypertension or advanced disease.

The patient was coinfected with HIV since 1993 (viral load at baseline was 118.000 copies/mL) and his CD4 T-cell count was 554 cell/μL. In 2000 he began antiretroviral therapy (ART) with different antiretroviral regimens associated to lamivudine to avoid HBV reactivation; furthermore, he was treated with methadone as a maintenance treatment. However, he was a poor adherent to ART therapy and consequently he did not reach a persistently complete virological suppression, even if he maintained a level of CD4 T-cells higher than 500 cells /μL. In May 2019, with the availability of pangenotypic DAA combinations he was treated for 12 weeks with a combination of sofosbuvir 400 mg/velpatasvir 100 mg.

No testing for resistance-associated variants (RAVs)—naturally existing nucleic acid substitutions and polymorphisms in the HCV viral genome—was done before the therapy.

His HCV viral load before the treatment was 5,319,000 IU/mL. After 12 weeks of SOF/VEL treatment HCV viral load was undetectable. After six months, he achieved sustained virological response (SVR), with undetectable levels of HCV RNA. Unfortunately, in April 2021, he probably had a reinfection with an HCV viral load of 2,404,521 IU/mL. We carried out a careful re-evaluation of his genotype after the failure to obtain a sequence for genotype 4. Therefore, we examined the core and NS5B regions to determine the HCV genotype again, which resulted in 4d by sequencing the core region, but in 1b by sequencing the NS5B region. Then, the NS5A region was successfully sequenced using HCV genotype 1b-specific primers to examine the resistant-associated substitutions (RAS). The resistance assay showed 159F and 282R mutations conferring resistance to sofosbuvir.

From March 2022 onwards he was retreated for eight weeks with the new combination of glecaprevir 100 mg/pibrentasvir 400 mg according to the EASL (European Association for the Study of Liver) guidelines. Until May 2022, HCV viral load was undetectable.

We assumed that the patient might be infected with a recombinant HCV strain in consideration of the high risk of superinfection present in this category of patients. HCV core region and HCV NS3, NS5A and NS5B regions were amplified and sequenced. 

HCV RNA was extracted using the Qiamp viral RNA mini kit following the manufacturer’s instructions (Qiagen Viral RNA Mini Kit, Qiagen Italia, Milan, Italy). Then, production and amplification of cDNA was carried out in a single step using the Superscript III One-step RT-PCR system with Platinum Taq (ThermoFisher Scientific, Milan, Italy). Primers and amplification conditions used for genotyping and resistance assay were already described [13,14] Then, core, NS3, NS5A, and NS5B amplified products were purified and sequenced by an automated DNA sequencing analyzer (Applied Biosystems SeqStudio Genetic Analyzer, Milan, Italy) using the Big Dye Terminator Cycle Sequencing Kit v3.1 (ThermoFisher Scientific, Milan, Italy).

The sequences were aligned to the reference sequences belonging to genotype 4 and genotype 1 with different subtypes available in GenBank using the CLUSTAL X program. The only known recombinant strain 4/1e, identified for the first time in Cameroon, was also included as reference sequence. Neighbor-joining trees were generated using the maximum likelihood method with the Kimura 2 parameter using MEGA version 7.0. The reliability was estimated by 1000 bootstrap replications.

Putative recombinant sequences were identified with the SimPlot program [15], using Core, NS3, NS5A and NS5B sequences and the reference sequences obtained from the HCV sequence database. 

Phylogenetic analyses showed that the core region from the patient formed a monophyletic cluster with different subtypes belonging to genotype 4 including the recombinant 4/1e strain, while NS3, NS5A and NS5B clustered with HCV genotype 1 subtype b (Figure 1A–D).

Recombination analysis indicated a recombination breakpoint in the NS3 region around position 470 nt (Figure 2).

## 3. Discussion

Here we report the detection of a rare natural occurring HCV recombinant intergenotypic strain, 4/1b. To the best of our knowledge, this recombination is reported for the first time worldwide, even if a recombination between genotype 4 and genotype 1e was identified in two patients in Cameroon [16].

Recombination strains in HCV are underestimated because the routinely genotyping assays used are not able to detect recombinant forms. However, the understanding of this phenomenon in HCV is of great interest. In fact, recombination might determine a better viral fitness produced by the emergence of mutations able to provide drug resistance or escape from the host immune system [17,18,19]. Furthermore, there is limited knowledge of how these chimeric strains respond to therapy; for example, it was shown that the 2k/1b recombinant HCV strain is less responsive to antiviral therapy [20]. Recombination breakpoints for intergenotypic strains are often found in the NS2 region or between the NS2 and NS3 regions. In our patient, the recombination crossover point was estimated at nt 470 of the NS3 region. It is difficult to establish whether our patient had received intergenotypic recombinant HCV or if such recombination occurred because he was infected with both genotype 4 and 1b HCV strains.

The patient experienced a new positive HCV viral load with failure to respond to sofosbuvir/velpatasvir combination therapy. 

He has a number of risk factors that might contribute towards treatment failure: being a drug intravenous user makes him more susceptible to reinfection, being co-infected with other blood-borne viruses and being infected with a genotype 4. This genotype was rare in Europe, but its prevalence is increasing [21] and is characterized by high heterogeneity with more than 18 recognized subtypes [21,22,23]. In our case, we were not able to identify accurately the subtype to which the parent strain belonged to.

It has been shown that DAA regimens are highly efficient against genotype 4, even if numerous treatment failures are reported in patients infected with this genotype, due to emergent resistance mutations [24,25].

In our patient, we have detected mutations in the NS5B region associated to sofosbuvir resistance. 

Generally, a resistance assessment before the beginning of treatment is not recommended for naïve patients, while it might be required in previously treated patients who have failed to respond to DAA treatment, like in our case. For this reason, we were not able to establish whether the patient already harbored the recombinant strain at the start of the DAA therapy.

## 4. Conclusions

Identifying the HCV genotype/subtype and possibly recombinant strains might also be useful at the time of pangenotypic treatment to optimize the treatment regimens and to detect the presence of eventual baseline resistance patterns.

Thus, increasing the knowledge about viral resistance to DAA and the role of new recombinant strains on the efficacy of HCV treatment will be of great relevance to avoid future treatment failure.

## Figures and Tables

**Figure 1 ijerph-19-11655-f001:**
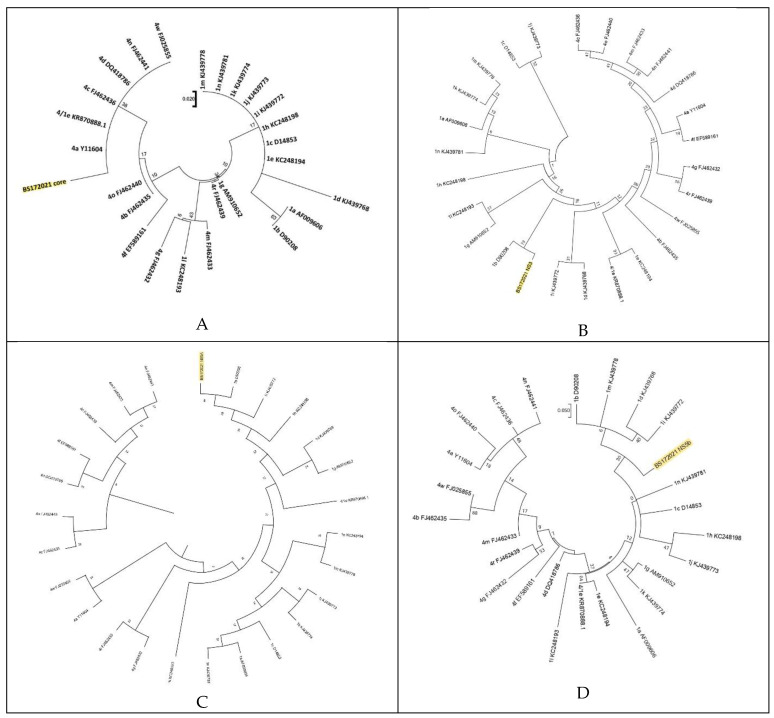
Phylogenetic tree analysis of a hepatitis C virus strain (BS172021) isolated from our patient compared with reference strains. The neighbor-joining tree was based on the core (**A**), NS3 (**B**), NS5A (**C**) and NS5B (**D**) sequences and the genetic distance was estimated using the Kimura 2 parameters model. The sequences identified in this study are marked in yellow. Reference sequences are labelled with their subtypes and accession numbers.

**Figure 2 ijerph-19-11655-f002:**
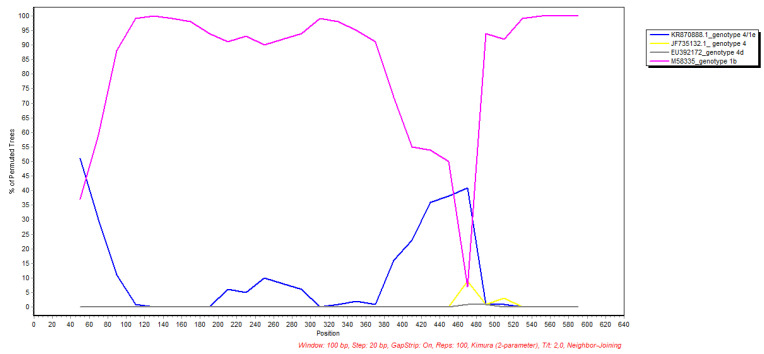
Bootscan plot of the NS3 region of our strain (BS172021) generated using the SimPlot software. It displays the bootstrap support for the clustering of the BS172021 NS3 region with 3 genotype reference genomes in 100-nt-sliding windows, 10 nt increments, and the Kimura 2-parameter model method with a transition-transversion (Ts/Tv) ratio of 2.0 across the viral genome. The curves are colored in yellow, blue, pink, and grey according to the genotype key on the right.

## Data Availability

The sequences obtained and described in this study are available in GenBank database (ON951660, ON951661, ON951662, ON951663).

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
