# Peer review of "DAA Treatment Failure in a HIV/HBV/HCV Co-Infected Patient Carrying a Chimeric HCV Genotype 4/1b"

_ijerph, 2022, doi:10.3390/ijerph191811655_

Round 1
Reviewer 1 Report
This paper presents a case report of pangenotypic direct antiviral agent (DAA) treatment failure in a HIV/HBV/HCV coinfected patient. Through sequencing of NS5a, NS5b, NS3 and core region the authors determined that the treatment failure was due to presence of recombinant HCV genotype 4/1b.
I have the following comments for the authors:
The figures in the manuscript are of poor quality and figure texts are difficult to read.
- Authors should include high resolution figures.
- Text in the figure 1 (A to D) showing phylogenetic tree analysis is not readable. Font should be made bigger.
- Fonts of the figure 2 plots should be made bigger to make it readable.
Authors write that HCV core region and HCV NS3, NS5A, and NS5B regions amplified and sequenced. The sequences should be included with the manuscript. Authors should also provide details on method of isolation and sequencing.
The patient was coinfected with HBV and HIV. The authors should discuss whether these could potentially contribute to treatment failure.
Line 139: the authors make a claim but don’t provide any reference.
Author Response
Answers to Reviewer 1
Comments and Suggestions for Authors
This paper presents a case report of pangenotypic direct antiviral agent (DAA) treatment failure in a HIV/HBV/HCV coinfected patient. Through sequencing of NS5a, NS5b, NS3 and core region the authors determined that the treatment failure was due to presence of recombinant HCV genotype 4/1b.
I have the following comments for the authors:
The figures in the manuscript are of poor quality and figure texts are difficult to read.
- Authors should include high resolution figures.
- Text in the figure 1 (A to D) showing phylogenetic tree analysis is not readable. Font should be made bigger.
- Fonts of the figure 2 plots should be made bigger to make it readable.
- All the figures have changed in 400 dpi to increase the resolution and have sent to the journal as JPEG files.
Authors write that HCV core region and HCV NS3, NS5A, and NS5B regions amplified and sequenced. The sequences should be included with the manuscript. Authors should also provide details on method of isolation and sequencing.
- The sequences were available in GenBank database and we have added this information in the Data availability statement. Details on amplification and sequencing methods have been added
The patient was coinfected with HBV and HIV. The authors should discuss whether these could potentially contribute to treatment failure.
- We agree with the reviewer. Coinfection with other viruses might contribute to the treatment failure and this has been indicated in the line 131
Line 139: the authors make a claim but don’t provide any reference.
- In the line 139, there is no claim but it was reported the result of our resistance assay by which we have detected the 159F and 282R mutations able to induce sofosbuvir resistance as indicated previous in the paper (lines 77-78)
Reviewer 2 Report
I find that the manuscript DAA treatment failure in a HIV/HBV/HCV co-infected patient 2 carrying a chimeric HCV genotype 4/1b Presents a very interesting case. There are a few suggestions for improvement; first, the English language must be reexamined by an English literature professor or a native speaker. Second, since the the virus vas not sequenced the first time the patient received DAA treatment but the genotype was determined by using the reverse hybridization method, there is, from what I understand, no way of knowing whether the NS5 region was genotype 1b to begin with, maybe there was no reinfection. So I would like the authors to argue that in the discussion/conclusion section a bit clearer. Other than that, I find the manuscript suitable for publishing.
Author Response
Answers to reviewer 2
I find that the manuscript DAA treatment failure in a HIV/HBV/HCV co-infected patient 2 carrying a chimeric HCV genotype 4/1b Presents a very interesting case.
There are a few suggestions for improvement; first, the English language must be reexamined by an English literature professor or a native speaker.
- The paper has been revised for the English language
Second, since the the virus vas not sequenced the first time the patient received DAA treatment but the genotype was determined by using the reverse hybridization method, there is, from what I understand, no way of knowing whether the NS5 region was genotype 1b to begin with, maybe there was no reinfection. So I would like the authors to argue that in the discussion/conclusion section a bit clearer. Other than that, I find the manuscript suitable for publishing.
- We agree with the referee’s observation, and we have added the term “probably” for reinfection and a short sentence at the end of the Discussion section